# Migration and Adhesion of B-Lymphocytes to Specific Microenvironments in Mantle Cell Lymphoma: Interplay between Signaling Pathways and the Epigenetic Landscape

**DOI:** 10.3390/ijms22126247

**Published:** 2021-06-10

**Authors:** Laia Sadeghi, Anthony P. Wright

**Affiliations:** Department of Laboratory Medicine, Division of Biomedical and Cellular Medicine, Karolinska Institutet, 141 57 Stockholm, Sweden; anthony.wright@ki.se

**Keywords:** mantle cell lymphoma, tumor microenvironment, lymphoma trafficking, epigenetic status, gene expression, epigenetic landscape, signaling pathway, cytokine and chemokine, adhesion molecules

## Abstract

Lymphocyte migration to and sequestration in specific microenvironments plays a crucial role in their differentiation and survival. Lymphocyte trafficking and homing are tightly regulated by signaling pathways and is mediated by cytokines, chemokines, cytokine/chemokine receptors and adhesion molecules. The production of cytokines and chemokines is largely controlled by transcription factors in the context of a specific epigenetic landscape. These regulatory factors are strongly interconnected, and they influence the gene expression pattern in lymphocytes, promoting processes such as cell survival. The epigenetic status of the genome plays a key role in regulating gene expression during many key biological processes, and it is becoming more evident that dysregulation of epigenetic mechanisms contributes to cancer initiation, progression and drug resistance. Here, we review the signaling pathways that regulate lymphoma cell migration and adhesion with a focus on Mantle cell lymphoma and highlight the fundamental role of epigenetic mechanisms in integrating signals at the level of gene expression throughout the genome.

## 1. Introduction

The tumor microenvironment, which includes various types of cells, is a location where tumor cells continuously interact with surrounding normal cells and extracellular molecules. In B-cell lymphomas, interaction of tumor cells with stromal cells in the tumor microenvironment promotes tumor cell growth, proliferation, survival and drug resistance [1,2,3]. The interactions between malignant B-cells and tumor microenvironment components alter the gene expression profiles of tumor cells in order to favor the survival of malignant B-cells [4,5,6].

Mantle cell lymphoma (MCL) is a rare and largely incurable B-cell non-Hodgkin lymphoma (NHL) accounting for 5–10% of all lymphomas [7]. MCL is characterized by the chromosomal translocation t(11;14)(q13;q32), which brings the cyclin D1 gene (*CCND1*) under control of the immunoglobulin heavy chain (*IGHV*) enhancer. This results in cyclin D1 overexpression and leads to aberrant cell cycle progression [8,9]. Most patients diagnosed with MCL are at an advanced disease stage with extra-nodal involvement, including the bone marrow, spleen and gastrointestinal tract [10,11,12]. In addition to cyclin D1 dysregulation, overexpression of transcription factor SOX11 has been observed in most conventional MCL cases [13,14,15]. More importantly, SOX11 is intensely expressed in cyclin D1-negative MCL cells and serves as a specific biomarker for the diagnosis of this subset of MCL [16]. Several studies have described other, secondary genomic alterations in MCL including mutations in the *ATM*, *TP53*, *MYC* and *NOTCH1* genes [17,18,19,20,21,22]. Comprehensive assessment of mutations in primary MCL cells using next-generation sequencing such as whole exome sequencing (WES), whole genome sequencing (WGS) and single nucleotide polymorphism array (SNP) have shown that MCL is a genetically heterogenous disease within individual patients as well as between different patients [23,24]. These studies demonstrated several recurrent mutations, including *ATM*, *TP53* and *MLL2,* but they also uncovered a number of novel and infrequent mutations in patients, which suggests divergent clonal evolution pattern in MCL cells [24]. As tumor cells divide, mutations occur randomly, resulting in multiple sub-clonal populations with different phenotypes within the same tumor, known as intra-tumor heterogeneity, which makes tumor more prone to recurrence and drug resistance.

Distinct from mutations, altered epigenetic status, i.e., chromosomal conditions that influence gene expression without changing the primary DNA sequence, plays an important role in the development of B-cell lymphomas [25,26,27]. Variable epigenetic components throughout the genome that influence gene expression include DNA methylation and patterns of histone modification, which alter chromatin structure, DNA accessibility and gene expression patterns [28]. The combination of genetic mutations and epigenetic changes induced by internal and external factors leads to constitutive activation of proto-oncogenes and the loss of tumor-suppressor gene activity that ultimately causes tumorigenesis [25,29,30]. Comprehensive methylation profiling by genome-wide array, comparing DNA methylation changes in MCL patients with normal B-cells, has demonstrated highly heterogenous methylation profiles across MCL patient samples [31]. This suggests extensive intra-tumor epigenetic heterogeneity in MCL cells. Thus, epigenetic disorders, together with genetic abnormalities, may contribute to different routes of clonal evolution, causing tumor development, therapy resistance and relapse. The broader context of genetic mutations associated with the development of NHLs has been extensively reviewed elsewhere [32,33,34] and is deemed to be outside the scope of this review.

In addition to the role of intrinsic abnormalities in tumor cells, an essential role of extrinsic factors, such as the tumor microenvironment, in tumor growth and development of drug resistance in B-cell lymphomas is widely accepted [35,36,37]. Even though the exact cellular composition of the MCL microenvironment is not yet clear, MCL cells in secondary lymphoid organs are known to interact with CD4^+^ and CD8^+^ T-cells, macrophages and mesenchymal stroma cells [38,39,40]. Interactions between tumor cells and microenvironment components trigger activation of a cascade of signals that travel to the nucleus, integrate with the specific epigenetic landscape of the tumor cell and thereafter cause specific alterations in gene expression. Adhesion of MCL cells to stromal cells in the tumor microenvironment causes activation of multiple signaling pathways, including the B-cell receptor (BCR), PCLγ2, PI3K/AKT, NF-κB and JAK/STAT signaling pathways [41,42,43]. These activated signaling pathways play a critical role in the inhibition of apoptosis, maintaining proliferation, survival and migration of MCL cells [6,44,45]. Crosstalk between tumor cells and microenvironment components promotes anti-tumor immune responses in B-cell lymphomas [46]. Interaction between CD40 on MCL cells and the CD40 ligand on the surface of T-cells in the microenvironment promotes tumor cell proliferation and induces PD-L1 (program-death ligand-1) expression on MCL cells [47,48]. Overexpression of PD-L1 on malignant B-cells enhances the interactions between PD-L1 and PD1, expressed on the surface of tumor-infiltrating T-cells, which results in inhibition of T-cell proliferation and causes T-cell exhaustion, which in turn promotes the immune escape of tumor cells [49,50]. The expression levels of PD-L1 are mainly regulated by microenvironment-mediated signaling pathways, which in part are controlled by epigenetic mechanisms [47,51,52,53]. Although the role of the tumor microenvironment in the regulation of cellular transduction pathways in tumor cells has already been described [6,54,55], much less is known about the interaction between cell signaling events and the specific epigenetic landscape of the cells that causes cell-type-specific effects on chromatin structure and thereby gene expression.

Here, we review current evidence indicating that epigenetic alterations contribute to the development of NHLs including MCL. Recent investigations have identified mutations in epigenetic regulators that cause epigenetic changes and thereby lead to abnormal gene activity in lymphocytes [30,56,57]. However, epigenetic alterations in lymphocytes have also been observed in the absence of mutations in genes encoding proteins important for epigenetic regulation, suggesting that the tumor microenvironment can induce epigenetic dysregulation in lymphoma cells. Thus, here we present some examples of the relationship between important cellular signaling pathways and chromatin-associated proteins that play crucial roles in the development and survival of B-cell lymphomas. We further propose a possible model where somatic mutations and the tumor microenvironment interact in order to shape the epigenetic landscape of lymphoma cells, which induces transcriptional changes in tumor cells, enabling their proliferation and survival.

## 2. Epigenetic Alteration in MCL

In eukaryotic cells, DNA is wrapped around histone proteins to form nucleosomes, the basic structural units of chromatin [58]. Chromatin structure and organization play a crucial role in regulating transcription by modulating DNA accessibility to the transcriptional machinery and transcription factors. Chromatin is organized into distinct structural and functional domains, termed euchromatin and heterochromatin, based on different combinations of regulatory proteins [59]. The establishment and maintenance of distinct chromatin domains play an important role by establishing the pre-conditions that determine the extent to which individual genes can be regulated by transcription factors [59]. Multiple mechanisms affecting chromatin organization, including DNA methylation, ATP-dependent chromatin remodeling and post-translational modification of histones, are required to ensure proper cellular functions. Different modifications of histone proteins are associated with discrete chromatin states that may be activated or repressed and impact a wide variety of cellular processes including transcription, DNA replication and DNA repair [60].

Currently known disease-causing epigenetic changes in B-cell lymphomas are predominantly induced by somatic mutations in epigenetic regulators, which have been identified by genome-wide screens and include key players involved in DNA methylation, histone modification and chromatin organization [32,57,61,62]. Here, we will briefly discuss mutations that modulate the epigenetic landscape of MCL as well as some other lymphomas, resulting in an abnormal transcriptional programming (Figure 1). A list of mutations known to be involved in epigenetic alterations in MCL is presented in Table 1.

### 2.1. DNA Methylation

DNA methylation mediates gene silencing by inducing a repressive chromatin structure, and it is involved in development, X-chromosome inactivation, genome stability and cell differentiation [82,83]. Extensive perturbation of cytosine methylation patterns is a hallmark of B-cell malignancies and occurs in specific patterns that can be distinguished in different subtypes of NHLs [63]. Analysis of the genome-wide DNA methylation profile in MCL patients revealed significantly different promoter methylation patterns between MCL patient samples and normal B-cells [63]. The different DNA methylation patterns were shown to contribute to the regulation of gene expression associated with lymphoma development. Queirós et al. identified hypomethylated regions downstream of SOX11 in a group of MCL cases, which suggests that epigenetic mechanisms may be involved in regulation of SOX11 in MCL [63]. This is in agreement with results from Li XY et al. who showed that DNA methyltransferase-1 (DNMT-1) is upregulated in MCL, causing global disruption of DNA methylation patterns [64]. Arsenic trioxide (As2O3, ATO), a DNA methyltransferase inhibitor, effectively decreased DNMT-1 mRNA expression levels and promoted apoptosis in MCL cells [64]. A further study using microarray-based gene expression profiling showed that promoter regions of tumor suppressor genes are hypermethylated in MCL cells, which coincided with their transcriptional repression [65]. In general, abnormal DNA methylation of tumor suppressor genes suppresses their transcription levels, and this is a common phenomenon observed in cancer.

### 2.2. Chromatin Remodeling

Nucleosomes play an important role in regulating higher-order chromatin structure. Chromatin remodeling enzymes use ATP to alter the structure and/or position of nucleosomes and thereby regulate the accessibility of DNA to transcription factors [84]. In eukaryotic cells, a wide variety of chromatin-remodeling complexes are present to remodel chromatin via different mechanisms [85]. For example, the SWI/SNF family of chromatin remodeling enzymes globally moves or evicts nucleosomes in order to create open chromatin regions, leading to transcriptional activation of genes throughout the genome [86].

Mutations in genes encoding proteins involved in chromatin remodeling have been identified in different types of cancer. SMARCA4, a central component of the SWI/SNF chromatin-remodeling complex, regulates transcription via remodeling nucleosome positions. Mutations affecting *SMARCA4* have been reported in both untreated and relapse/refractory MCL patients [66,67]. MCL patients with a *SMARCA4* mutation are resistant to combined therapy with Ibrutinib (BTK inhibitor) and Venetoclax (BCL-2 inhibitor) [66,67]. The mutation in *SMARCA4* affects chromatin accessibility and reduces expression of the transcription factor, ATF3, which negatively regulates transcription of the anti-apoptotic gene *BCL-xL*, leading to the observed resistance of MCL cells to therapy [66].

### 2.3. Histone Deacetylation

In general, histone acetylation is associated with active transcription. Acetylation of lysine residues on histone proteins increases DNA accessibility to transcription factors [87]. The lysine histone acetyltransferase (HAT) P300 and (CREB-binding protein) CBP, known as (P300/CBP), acetylates histone and non-histone proteins in order to activate transcription [87]. In contrast, histone deacetylases (HDACs) remove acetyl groups from hyperacetylated histones in order to suppress gene transcription [88]. In addition to histone deacetylation, HDACs also regulate the acetylation status of a variety of non-histone proteins [89]. The balance between acetylation and deacetylation is important for normal gene expression patterns as it, e.g., helps to define euchromatin (active) and heterochromatin (repressed) regions throughout the genome. In tumor cells, this balance is often disrupted due to aberrant HDAC levels that lead to inactivation of tumor-suppressing genes. Elevated expression of HDACs has been observed in several cancer types including NHLs and knockdown of HDACs, resulting in increased apoptosis and cell cycle arrest [90,91].

HDAC inhibitors (HDACi) are a class of chemical compounds that increase histone acetylation and restore the balance between pro- and anti-apoptotic proteins and induce apoptosis in tumor cells. Several HDACi, including Vorinostat (SAHA), are in clinical trials for the treatment of leukemia and lymphoma [92]. HDACi have been used singly or in combination with other compounds, e.g., Rituximab (anti-CD20 antibody), to treat patients with MCL [93]. Inhibition of HDACs induced cell death in MCL cell lines by enhancing histone acetylation at the promoter region of pro-apoptotic genes including (*BIM* and *BMF*), which induced their transcription [68]. In addition, Vorinostat in combination with Palbociclib (selective inhibitor of cyclin-dependent kinase 4/6 involved in cell cycle progression) reduced MCL cell growth and induced apoptosis significantly by increasing histone H3 acetylation and inhibiting BCL-2 family members. Overexpression of BCL-2 family proteins attenuates apoptosis in tumor cells [69].

Another HDACi (PCI-24781) enhanced acetylation of histone H3 at the promoter region of the gene encoding P21 (*CDKN1A*), causing increased chromatin accessibility for binding of transcription factors [70]. Elevated levels of P21 mRNA induced G1 cell cycle arrest. Moreover, PCI-24781 significantly downregulated the expression of NF-κB subunits (NF-κB1 and REL B) and NF-κB target genes (*CLL3*, *CLL4*, *IL-6*, *MYC*) [70]. This suggests a role for histone deacetylation in the regulation of NF-κB activity, which remains poorly understood. A combination of PCI-24781 and bortezomib (a proteasome inhibitor) synergistically suppressed NF-κB and induced apoptosis in lymphoma cells [70].

HDAC3 has been implicated as a regulator of PD-L1 expression in B-cell lymphomas [94]. Inhibition of HDAC3 reversed microenvironment-mediated immunosuppression and resulted in better clinical response to PD-L1 blockage [94]. In Chronic lymphocytic leukemia (CLL), the combination of HDAC6 inhibitor (ACY-738) with PD-L1 blockage using monoclonal antibodies significantly increased anti-tumor efficiency of T-cells and reduced tumor burden [95]. Thus, the combination of HDAC inhibitors and checkpoint blockage can help to overcome immunotherapy resistance.

### 2.4. Histone Methylation

Post-translational modification of histone proteins by methylation is an important and common modification that influences many cellular processes. The major methylation sites on histone proteins are lysine and arginine residues and methylation of non-histone proteins is also frequently observed [96,97]. Depending on which residue is methylated and the degree of methylation, the effect of methylation on functional outcome can be different [96]. Histone methylation is a dynamic process, and methyl groups can be removed from histone proteins by histone demethylases [98]. Methylation of histone proteins provides a scaffold for the recruitment and assembly of other proteins involved in chromatin remodeling and transcription regulation [96]. Histone proteins can be mono-(me1), di-(me2) or tri-methylated (me3). The abnormal expression of various methyltransferases and demethylases has been reported in many tumor types, suggesting that the level and degree of histone methylation contributes to tumorigenesis and that the enzymes involved provide promising targets for anti-cancer treatment [79,99,100].

WHSC1 (also known as MMSET or NSD2) is a specific histone methyltransferase, responsible for histone H3 methylation at lysine 36, and it is mutated in a subgroup of B-cell malignancies including MCL [72,73]. The mutation enhances the methyltransferase activity of WHSC1, resulting in altered genome-wide H3K36me methylation and a concomitant global decrease in the levels of the H3K27me3 repressive mark in chromatin [101]. Alterations in histone methylation status can switch a gene from an active to a repressed state or vice versa. As a result of altered WHSC1 levels, the expression of genes positively involved in proliferation and cell-cycle progression increases, suggesting that *WHSC1* functions as an oncogene favoring tumor cell growth and proliferation.

*MLL2* encodes a highly conserved histone methyltransferase that mediates methylation of lysine 4 in histone H3 (H3K4) [102]. Trimethylated H3K4 (H3K4me3) is a conserved mark associated with active transcription, and active enhancers are marked by H3K4me1 [103]. Tri-methylation of histone H3 at lysine residues 4 and 27 (H3K4me3 and H3K27me3) on the promoter region plays a crucial role in the regulation of gene transcription. While H3K4me3 is associated with the activation of gene expression, H3K27me3 is a mark for repressed genes [104,105]. The promoter region is generally uniquely marked with either H3K4me3 or H3K27me3, except in the case of bivalent chromatin at the promoter region of developmental regulatory genes [104,105,106]. Bivalent chromatin is marked with histone modifications associated with both gene activation and repression and keeps repressed genes poised and ready for rapid activation [106]. Bernhart S et al., using ChIP-seq (chromatin immunoprecipitation combined with high-throughput sequencing), demonstrated that chromatin structure at bivalent promoters in cancer cells (both in solid and hematological tumors) is disrupted, resulting in deregulation of gene silencing and activation of poised genes [107].

*MLL2* has been identified as one of the commonly mutated genes in NHLs including Diffuse large B-cell lymphoma (DLBCL) and MCL [66,67,74]. In DLBCL, mutation of *MLL2* produces a truncated *MLL2* protein that lacks the SET domain (required for its methyltransferase activity) [74]. Although the role of *MLL2* in MCL progression is not well understood, *MLL2* loss-of-function mutations are known to diminish H3K4 methylation and drive tumor cell growth and proliferation, suggesting that *MLL2* functions as a tumor suppressor gene [75,108]. Interestingly, a recent study has provided evidence for the role of *MLL2* in maintaining genomic stability, since the loss of *MLL2* caused chromosomal aberrations and accumulation of mutations [76].

The multi-protein polycomb complexes (PRCs) promote transcriptional repression by disturbing nucleosome occupancy and mediating H3K27 methylation. H3K27 methylation is catalyzed by EZH2 in the presence of cofactors SUZ12 and EED. EZH2 is a component of PRC2 that catalyzes mono-, di- and trimethylation of lysine 27 of histone H3 (H3K27) [109,110,111]. All three forms of H3K27 methylation are mutually exclusive and are present at distinct genomic regions. Trimethylated H3K27 (H3K27me3) is present at promoter regions of silenced genes and is associated with transcriptional repression [112]. To preserve chromatin structure and prolong gene silencing, H3K27me3 recruits PRC complexes to nucleosomes of the nascent DNA strand during DNA replication in order to maintain H3K27me3 levels [113].

One of the most frequent genetic alterations observed in germinal center B-cell (GCB) lymphomas such as Follicular lymphoma (FL) and DLBCL is mutations of *EZH2* that affect the SET domain. *EZH2* is essential for normal B-cell development and rearrangement of the immunoglobulin heavy chain (*IGHV*) [114]. Moreover, *EZH2* is required for the differentiation of germinal B-cells into memory cells through the establishment and maintenance of bivalent modifications at the promoter region of key regulatory genes [115].

Gain-of-function mutation of *EZH2* at tyrosine 641 (Tyr641) in FL and DLBCL leads to enhanced *EZH2* stability, resulting in increased H3K27me3 levels [57,116]. In addition to mutation, other genetic lesions affecting *EZH2*, including chromosomal gain or loss and DNA hypermethylation, have been identified in B-cell malignancies [100,117]. *EZH2* mutation has not been identified in MCL patient samples or MCL cell lines [32,61], although PRC2/EZH2 complex overexpression has been reported in MCL cell lines [77,78]. The work of Demosthenous et al. showed that deregulation of PRC2/EZH2 in MCL cell lines mediates epigenetic silencing of the *CDKN2B* CDK inhibitor gene to promote MCL cell proliferation and survival [77]. In addition, it has been shown that EZH2 overexpression in MCL cell lines and primary cells facilitates recruitment of the DNA methylation machinery, enabling more stable and long-term repression of HOX genes [78]. Until now, genome-wide analysis of H3K27me3 distribution in MCL cells compared with normal B-cells has not been conducted. Given the role of dysregulated EZH2 in the initiation and progression of NHLs, several selective inhibitors have been developed in recent years [118]. However, EZH2 inhibitors failed to attenuate EZH2 levels in malignant B-cells due to unspecific mechanisms of action [118]. Further studies are required to develop strategies to specifically alter H3K27me3 levels in tumor cells.

### 2.5. Histone Demethylation

Post-translational modification of histone H3 at lysine 27 has an essential role in both transcriptional repression and activation. Studies have shown that both increase and decrease in the activity of enzymes regulating H3K27me3 levels are associated with carcinogenesis [61,80]. Despite the opposing activities of EZH2 and the KDM6B demethylase in affecting H3K27me3 levels, both these enzymes have been implicated in both lymphoma development and drug resistance.

Histone demethylase KDM6B/JMJD3 removes di- and tri-methyl marks from H3K27 and reverses PRC2/EZH2-mediated transcription repression [119,120]. Genome sequencing has not so far identified *KDM6B* mutations in NHLs, except for a chromosomal deletion in Sezary syndrome, an aggressive T-cell lymphoma [121]. In contrast, KDM6B overexpression has been observed in Hodgkin’s lymphomas (HLs), DLBCL, Multiple myeloma (MM) and Acute myeloid leukemia (AML) [79,80,122,123]. KDM6B is required for maintaining hematopoietic stem cell (HSC) self-renewal ability and repopulation capacity and plays an important role in the differentiation of memory B-cells [122,124].

In MM, KDM6B regulates the expression of key components of the MAP Kinase pathway in a demethylase-activity-independent manner in order to enhance tumor cell survival, whereas in DLBCL, KDM6B controls BCL6 expression and protein levels in a fashion that requires the demethylase activity in order to facilitate tumor cell proliferation [79,80]. KDM6B has been identified as a new therapeutic target in B-cell malignancies, and its selective inhibitor GSK-J4 has been shown to counteract cell proliferation in different tumor types such as AML, MM and DLBCL [80,123].

KDM6B overexpression can antagonize EZH2-mediated repression [125] of tumor suppressor genes in solid tumors by reducing H3K27me3 levels, while inhibition of KDM6B boosts apoptotic response to PI3K/AKT inhibitor treatment. This suggests that KDM6B can act as a tumor suppressor gene or an oncogene depending on cellular context [126].

## 3. Tumor-Microenvironment Mediated Epigenetic Alterations

As discussed above, cancer-associated mutations that affect the epigenetic machinery can lead to imbalances that change the epigenetic landscape in cancer cells and its effect on gene expression programs. However, the epigenetic landscape of cells is also susceptible to environmental cues associated with changes in normal or pathological contexts. For example, receptors and linked signal transduction pathways interpret microenvironment cues and signals and convert them into specific transcriptional states by modulating chromatin structure. Regulation of gene expression in response to signaling pathways requires post-translational modifications of histones and nucleosome rearrangement, which promotes chromatin reorganization, leading to activation or repression of transcription at particular loci throughout the genome. Interactions between malignant B-cells and stromal cells activate multiple signaling pathways, including PI3K/AKT and NF-κB, that function as signal transducers for cell-surface receptors, particularly receptor tyrosine kinases, in order to alter cellular function [6]. Here, we focus on some examples of key tumor microenvironment-mediated signaling pathways that have a direct impact on the chromatin state of tumor cells.

### 3.1. Role of the PI3K/AKT Pathway in Chromatin Modulation

The PI3K/AKT pathway plays a key role in many cellular mechanisms and is often dysregulated in malignant B-cells [127]. The PI3K/AKT pathway regulates essential cellular functions including growth, proliferation, metabolism, differentiation and motility in normal and malignant B-cells [128]. The PI3K catalytic subunit mediates phosphorylation of phosphatidylinositol-4,5-bisphosphate (PIP2) to generate phosphatidylinositol-3,4,5-trisphosphate (PIP3), which in turn acts as a second messenger to assemble and activate downstream signaling complexes, including protein kinase B, also known as AKT [129]. Phosphorylated AKT mediates phosphorylation of different proteins, including the mammalian target of rapamycin (mTOR), and the IκB inhibitor of NF-κB in order to regulate distinct cellular functions [127,130,131]. Moreover, PI3K/AKT phosphorylates chromatin-associated proteins and modulates chromatin structure (Figure 2).

Adhesion of MCL cells to stromal cells in the tumor microenvironment activates the PI3K pathway, which transmits signals from cell membrane receptors, such as the B-cell receptor and CXCR4 (chemokine receptor), to the nucleus, leading to transcriptional regulation of genes involved in migration, proliferation and survival [45,127,132,133]. Here, we highlight the effects of the PI3K/AKT on tumorigenesis via chromatin modifications that dramatically alter gene expression. Activation of PI3K/AKT by upstream signaling pathways mediates phosphorylation of the histone acetyltransferase (HAT), P300/CBP, which is responsible for acetylation of histone and non-histone proteins [134,135]. AKT-mediated phosphorylation of P300/CBP facilitates recruitment of the transcription machinery at the promoter region of genes involved in the survival of MCL and CLL cells [127,136]. MCL cells express high levels of CXCR4, a cell surface receptor required for trafficking and homing of normal and malignant B-cells [137]. Expression levels of CXCR4 are regulated by the transcription factor, FOXO1, and acetylation of histones by P300/CBP is a prerequisite for FOXO1/P300-mediated recruitment of transcription machinery at the promoter regions of important target genes [138,139]. PI3K inhibitor, Idelalisib has been used for the treatment of MCL, but the majority of patients eventually develop resistance to Idelaisib. Zhou XR et al. reported that a P300/CBP inhibitor (A-485) overcomes resistance to Idelalisib in MCL by reducing histone acetylation at the promoter region of receptor tyrosine kinase (RTK) genes, thereby, reducing their transcriptional upregulation by an independent mechanism [140].

Furthermore, activated PI3K/AKT phosphorylates and stabilizes DNA methyltransferase 1 (DNMT1) [141]. In mammals, DNA methylation is performed by DNMT1 and DNMT3, which are functionally and structurally distinct [142]. DNMT1 maintains the DNA methylation state [142]. In general, maintenance of DNA methylation activity is essential for the preservation of genomic integrity and inhibition of inappropriate gene activation. Abnormal activation of PI3K/AKT might contribute to tumor cell growth by favoring DNA hypermethylation and chromatin compaction at the promoter region of specific target genes, thereby suppressing their expression [141]. Accumulating evidence demonstrates a role for DNMT1 in malignant progression of NHLs including DLBCL and MCL by enhancing cell cycle progression [64,143,144].

In addition, PI3K/AKT is involved in regulating histone methylation levels. AKT interacts with and phosphorylates EZH2 at Serine 21 and suppresses its methyltransferase activity, which results in reduced H3K27me3 levels at the promoter regions of target genes, thus disrupting gene silencing [145]. A Serine to Alanine substitution at the phosphorylation site increases EZH2 association with H3 [145].

Tumor microenvironment-mediated upregulation of EZH2 has been reported in various types of NHL [146,147]. In a subset of CLL cells, signals from the microenvironment result in pronounced upregulation of EZH2 mRNA and protein levels, which induces anti-apoptotic responses and enhances cell viability [146]. Moreover, dysregulation of EZH2 levels in CLL results in intra-tumor heterogeneity contributing to drug resistance [148].

### 3.2. NF-κB and Chromatin Remodeling

The DNA binding activity of the nuclear factor-κB (NF-κB) transcription factor can be induced by an extracellular stimulus [149]. NF-κB is one of the key transcription factors that are expressed in almost all cell types and tissues. NF-κB binds to the promoter and/or enhancer region of its target genes, which are involved in stress responses, cell proliferation and apoptosis, and regulates their transcription [150]. The NF–κB pathway plays a crucial role in normal B-cell development and survival. Dysregulation of the NF-κB pathway blocks B-cell development and reduces the number of mature B-cells [151]. Receptor-mediated activation of NF-κB transcription factor subunits (RELA/P65, RELB, c-REL, p105/p50 and p100/p52) triggers their translocation to the nucleus, where they regulate the expression of NF-κB target genes that encode proteins such as cytokines, chemokines and adhesion molecules [152]. The NF-κB signaling pathway is divided into separate canonical and non-canonical pathways that regulate distinct gene expression programs. However, a crosstalk between the two pathways can potentially be mediated via NIK (NF-κB-inducing kinase) and CD40 [153]. Disruption of the NF-κB signaling pathway has been identified as one of the main drivers in the pathogenesis of several B-cell malignancies, including MCL [6,55].

Activation of NF-κB in response to cytokines and chemokines in tumor microenvironments promotes upregulation of EZH2 in leukemia and lymphoma cell lines [154]. Natoli G et al. showed that exposure of macrophages to different inflammatory cytokines elevated KDM6B/JMJD3 expression (H3K27me specific demethylase) through activation of the NF-κB pathway [81]. In a recent study from our lab, we showed that stromal cell adhesion mediates overexpression of KDM6B in MCL cell lines, which could potentially lead to the removal of the H3K27me3 mark from the promoter regions of genes encoding NF-κB subunits and thus activate their expression and nuclear localization. Enhanced NF-κB activity mediates transcription of migration and adhesion-related genes and promotes tumor cell survival.

## 4. Targeting PI3K in MCL

Given the crucial role of PI3K/AKT in lymphoma development and survival, different compounds have been identified that inhibit the PI3K activity. Idelalisib (formerly known as GS-1101 and CAL-101), the first class of PI3K inhibitors, has been used as a single-agent monotherapy or in combination with other inhibitors/agents to treat refractory or relapsed MCL and CLL patients [155]. Idelalisib inhibits MCL and CLL cell migration toward CXCL12 gradients, promotes egression of malignant B-cells from lymph nodes and induces apoptosis of malignant B-cells [155,156,157].

Combined PI3K and HDAC inhibitors have demonstrated significant anti-tumor effects in MCL and DLBCL [71,158]. CUDC-907 (dual PI3K and HDAC inhibitor) inhibits growth and proliferation of both Ibrutinib-sensitive and Ibrutinib-resistant MCL primary cells in vitro and induces apoptosis and cell cycle arrest through downregulation of PLK1, polo-like kinase 1, a key regulator of mitosis [71]. Moreover, CUDC-907 downregulated MYC in DLBCL and showed a promising anti-tumor effect [158]. Given the role of PI3K/AKT in priming chromatin for transcriptional activation, inhibiting its kinase activity can cause transcriptional repression of oncogenes, such as MYC [133]. Interestingly, Idelalisib in combination with Ibrutinib (BTK inhibitor) exhibits a stronger effect on MCL or CLL cell adhesion compared with either compound alone, indicating that Ibrutinib and Idelalisib synergistically target cytokine and integrin-mediated MCL and CLL cell adhesion [157].

## 5. Conclusions

Currently, chemotherapeutic treatment approaches, such as R-CHOP (rituximab, cyclophosphamide, doxorubicin, vincristine and prednisone) and targeted treatments such as Ibrutinib (BTK inhibitor) and Venetoclax (BCL2 inhibitor) are used to treat MCL. Despite effective treatment, MCL often recurs, partially due to drug resistance caused by alterations in the gene expression profile of tumor cells that survive treatment. These gene expression changes result from changes in the epigenetic landscape of residual malignant cells that result from mutations and/or microenvironmental cues. Thus, cancer is a complicated disease caused by the accumulation and combination of genetic and microenvironmental effects that together modulate the epigenome and ultimately cause abnormal gene expression. Thus, in line with recent research, genetic and epigenetic mechanisms are interconnected and should not be seen as separate mechanisms in cancer development. Epigenetic alterations can suppress gene activity and cause loss of function, and genetic mutation in an epigenetic regulator can lead to such epigenetic changes. Thus, understanding the relationships between changes in the genome and environmental cues (e.g., from microenvironments) and how they influence the epigenetic landscape is of fundamental importance for understanding mechanisms that drive cancer and the design of more efficient therapies. Since epigenetic mechanisms have central role in defining cell-type-specific gene expression, a single mutation affecting such mechanisms can cause genome-wide transcriptional dysregulation that affects multiple signaling pathways, leading to an altered cell type (e.g., change from normal to malignant). Such new cell types will be selected and prosper if they can compete with normal cells in key environmental niches (microenvironments). Equally, activation of signaling pathways in response to altered environmental cues can alter the epigenetic landscape and lead to the creation of new cell types. Thus, input from signaling pathways is integrated by epigenetic regulators in order to establish a cell-type-specific transcriptional program in normal and malignant cells.

In this review, we have presented examples of how the microenvironment modulates the epigenetic landscape in tumor cells through signaling pathways and changes their transcriptional program in order to promote tumor cell survival and proliferation at the expense of normal cells. It is important to note that the epigenome of cells is dynamic over time as it continuously integrates cues from the cellular environment with intrinsic and more stable cues that define cell type. As such, epigenetic landscapes associated with malignant cells are intrinsically reversible and therefore represent potential as targets for drug therapy. The relatively common occurrence of cancer-associated mutations in components of the epigenetic machinery seems at first glance to be a good strategy (from the point of view of the malignant cell) for irreversibly changing the epigenetic landscape in order to stably favor the selection and expansion of cells with malignant characteristics. However, the common occurrence of enzyme systems with opposite functions (acetylase/deacetylase, methylase/demethylase) suggests that malignant effects resulting, e.g., from loss-of-function mutations in a methylase could potentially be redressed by an inhibitory drug that targets an appropriate demethylase. Thus, identifying the molecular mechanisms and signaling pathways that define the epigenetic landscapes of tumor cells is an exciting field of research that should be addressed in order to improve future treatment options.

MCL is a heterogenous disease. In recent years, advances in massively sequencing technologies have defined the degree of genetic and epigenetic heterogeneity not only among individuals with MCL but also within individual patients. Understanding of genetic and epigenetic heterogeneity can improve therapeutic outcomes by directly targeting intra-tumor heterogeneity. However, current knowledge about intra-tumor heterogeneity is limited because most studies are based on bulk analysis of heterogenous cell populations. Extrinsic factors, such as tumor microenvironments, probably contribute to epigenetic heterogeneity, since homogenous cells may be expected to respond differently to different microenvironmental cues within microenvironments. The composition of cells in the tumor microenvironment, microenvironment-associated signaling pathways and the relative spatial localization of cells in microenvironments would all be expected to play a major role in generating intra-tumor heterogeneity. Technologies like single-cell analysis will increasingly allow the investigation of genetic and epigenetic factors that cause intra-tumor heterogeneity in order to further our understanding of clonal evolution and to develop better treatment strategies.

## Figures and Tables

**Figure 1 ijms-22-06247-f001:**
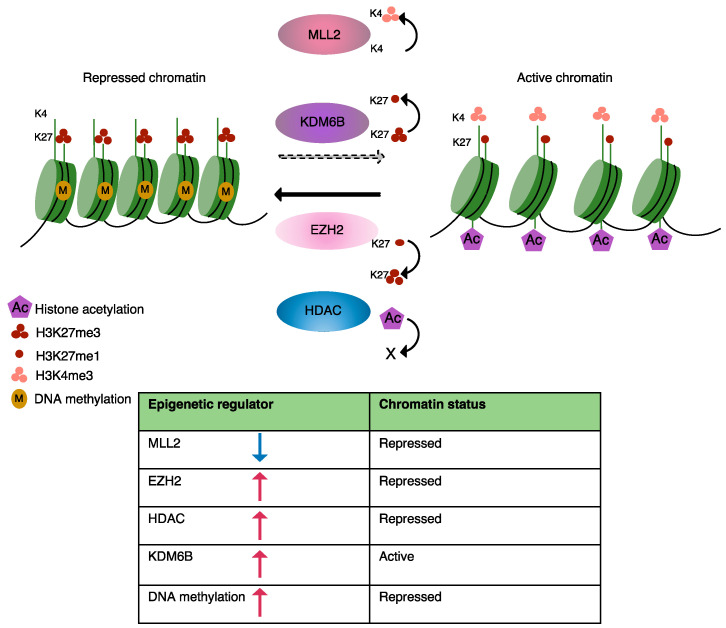
Schematic illustration, representing the role of histone-modifying enzymes and DNA methylation in regulation of chromatin structure in normal and tumor cells. EZH2 (light pink, PRC2 complex subunit) catalyzes tri-methylation of lysine 27 on histone H3 to repress gene expression. HDAC (blue) removes acetyl groups from histone proteins to reduce DNA accessibility to transcription machinery. KDM6B (purple) removes di- and trimethylated marks on histone H3 and MLL2 (dark pink) puts a trimethylation mark on histone H3 to increase DNA accessibility for transcription factors. In lymphoma, cell mutations affecting epigenic regulators alter the epigenetic landscape and change gene expression.

**Figure 2 ijms-22-06247-f002:**
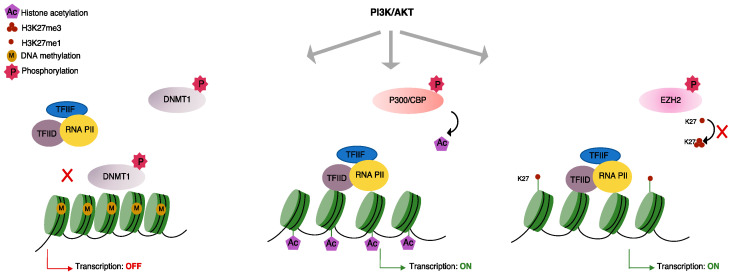
PI3K mediates phosphorylation of chromatin-associated proteins, affecting the epigenetic landscape and thereby expression of target genes. Activated PI3K/AKT phosphorylates and stabilizes DNA methyltransferase 1 (DNMT1) at the promoter region of target genes, which decreases the rate of transcription. PI3K/Akt mediates phosphorylation of P300/CBP and facilitates recruitment of transcription machinery at the promoter region of target genes, which increases the rate of transcription. AKT phosphorylates EZH2 and suppresses its methyltransferase activity, which results in reduced H3K27me3 levels at the promoter region of target genes and disrupts gene silencing.

**Table 1 ijms-22-06247-t001:** A summary of dysregulated epigenetic mechanisms associated with altered gene expression in MCL.

Dysregulated Epigenetic Mark in MCL	Gene/Genes Mediating Epigenetic Dysregulation in MCL	Type of Dysregulation	Target Gene/Genes Affected by Epigenetic Dysregulation	Putative Role in MCL	Reference
DNA methylation	-	DNA hypomethylation at promoter region of target gens	Mediating the expression of *SOX11*	Promotes oncogenic cell proliferation	[63]
DNA methylation	DNMT1 upregulation	Global DNA hypermethylation	Abnormal expression of β-catenin which upregulates the expression of *c-MYC* and *MMP7* (Matrix Metallopeptidase 7) Reduced expression of tumor suppressor gene; *PARG1*	Promotes tumor cell proliferation and survival	[64,65]
Histone H3K27acetylation	Mutation in the component of SWI-SNF complex (*SMARCA4*)	Abnormal histone H3K27 acetylation and chromatin accessibility at the promoter/enhancer region of target genes	Reduced chromatin accessibility at promoter region of transcription factor *ATF3* (negative regulator of anti-apoptotic gene *BCL-xL*)	MCL cell survival and drug resistance	[32,66,67]
Global histone acetylation	Abnormal activity of HDACs, Class I, II, e.g., HDAC8	Enhanced HDAC (Histone deacetylase) activity leading to abnormal histone acetylation and chromatin accessibility	Reduced transcription of pro-apoptotic genes (*BIM*, *BMF*) Enhanced expression of *c-MYC* and *PLK1*	Inhibits apoptosis and promotes tumor cell proliferation and survival	[68,69,70,71]
Histone H3K36me3	Gain of function mutation in histone methyltransferase *WHSC1* (*MMSET*)	Enhanced H3K36me3 levels	Enhanced expression of cell cycle regulators	Promotes tumor cell proliferation	[72,73]
Histone H3K4 methylation	Loss of function mutation in histone methyltransferase *MLL2 (KMT2D)*	Diminished H3K4 methylation levels	Functional consequences in MCL are not well understood Contributes to genome instability and transcriptional stress	Disturbs the expression of genes that sustain proliferation and cell survival	[32,74,75,76]
H3K27me3	EZH2 upregulation	Enhanced H3K27me3 levels	Repressed expression of *CDKN2B*, *HOX* genes	Promotes MCL cell growth	[77,78]
H3K27me3	KDM6B histone demethylase	Enhanced KDM6B levels, reduces H3K27me3 at promoter region of target genes	Functional consequences in MCL are not well understood In other B-cell malignancies target NF-κB subunits and target genes	Promoters tumor cells survival and drug resistance	[79,80,81]

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
