# Peer review of "Migration and Adhesion of B-Lymphocytes to Specific Microenvironments in Mantle Cell Lymphoma: Interplay between Signaling Pathways and the Epigenetic Landscape"

_ijms, 2021, doi:10.3390/ijms22126247_

Round 1

Reviewer 1 Report

This is a well organized, concise review article regarding tumor microenvironment, epigenetic alterations and signaling pathways in mantle cell lymphoma.  The manuscript provides updated information in the field. However, Table 1 is less informative and more details are needed, such as mutation pattern, frequency and biology in lymphoma cells.

Author Response

This is a well organized, concise review article regarding tumor microenvironment, epigenetic alterations and signaling pathways in mantle cell lymphoma.  The manuscript provides updated information in the field. However, Table 1 is less informative, and more details are needed, such as mutation pattern, frequency and biology in lymphoma cells.

Response: We agree with the reviewer and Table 1 is now modified. While data on mutation frequency was not clear from the source publications, we have added three more columns to explain the mutations influencing epigenetic state in MCL, changes in epigenetic state in MCL compared with normal cells and the role of dysregulated epigenetic mark in MCL progression.

Reviewer 2 Report

The authors reviewed the interplay between signaling pathways and the epigenetic landscape. While the section on epigenetic alterations in mantle cell lymphoma is offering a lot of information, the section on the tumor-microenvironment mediated epigenetic alterations is underdeveloped.

Some items to improve are listed below.

Major points:

The tumor microenvironment contains different cell types, including lymphoma cells and immune cells. Epigenetic alterations could affect the cross-talk between the cancer cell and the stromal cells or directly affect the supportive immune cells. Epigenetic changes can affect genes encoding tumor suppressors, inhibitory cytokines, and immune checkpoint molecules, e.g., PD-L1, leading to impaired anti-cancer immunity. Epigenetic modifications can also be found in tumor-associated immune cells, including myeloid cells, CD4+ T cells, and CD8+ T cells. The authors can improve the microenvironment section by highlighting the importance of epigenetics changes in the supportive immune cells and how combination therapy could influence this.

The role of the PI3K/AKT pathway or NF-kB in chromatin remodeling is only supported by indirect evidence. For example, are there manuscripts published showing chromatin remodeling (ATAC-seq) or H3K27me (ChIP-seq) in MCL cells after co-culture or idelalisib treatment?

Tumor heterogeneity, especially intratumor heterogeneity, is one of the major hallmarks of cancer, especially in blood cancer. Within the tumor-microenvironment, there is diversity in the phenotypes of tumor cells and the infiltration and differentiation status of immune cells. Therefore, looking at the bulk of cells doesn’t give a complete picture. The authors could provide more details on the heterogeneity of MCL and how current studies on epigenetic alterations failed to capture the heterogeneity.

Minor points:

The SMACA4 mutation (137-144) was found in relapsed/refractory MCL patients

It is unclear which cancer is discussed in sentences 215-218.

What is the role of MLL2 mutation in MCL? (222-226)

It is unclear which cancer is discussed in sentences 282-285.

Author Response

Reviewer 2

The tumor microenvironment contains different cell types, including lymphoma cells and immune cells. Epigenetic alterations could affect the cross-talk between the cancer cell and the stromal cells or directly affect the supportive immune cells. Epigenetic changes can affect genes encoding tumor suppressors, inhibitory cytokines, and immune checkpoint molecules, e.g., PD-L1, leading to impaired anti-cancer immunity. Epigenetic modifications can also be found in tumor-associated immune cells, including myeloid cells, CD4+ T cells, and CD8+ T cells. The authors can improve the microenvironment section by highlighting the importance of epigenetics changes in the supportive immune cells and how combination therapy could influence this.

Response: We thank the reviewer for this insightful comment. We have added new paragraphs of text to the paper regarding the role of microenvironment in anti-tumor immune responses (73-76), (83-93), and (207-213).

The role of the PI3K/AKT pathway or NF-kB in chromatin remodeling is only supported by indirect evidence. For example, are there manuscripts published showing chromatin remodeling (ATAC-seq) or H3K27me (ChIP-seq) in MCL cells after co-culture or idelalisib treatment?

Response: There is a lack of genome-wide epigenetic studies in Mantle cell lymphoma. H3K27me3 levels have been compared between normal naïve and germinal center B-cells PMID: 24682267. Text has been modified (288-290)

Tumor heterogeneity, especially intratumor heterogeneity, is one of the major hallmarks of cancer, especially in blood cancer. Within the tumor-microenvironment, there is diversity in the phenotypes of tumor cells and the infiltration and differentiation status of immune cells. Therefore, looking at the bulk of cells doesn’t give a complete picture. The authors could provide more details on the heterogeneity of MCL and how current studies on epigenetic alterations failed to capture the heterogeneity.

Response: We agree with reviewer that the microenvironment influences intra-tumor heterogeneity. We have added new paragraphs of text to the paper to address intra-tumor heterogeneity in MCL along the lines suggested (44-54), (66-68), (507-521).

Minor points:

The SMACA4 mutation (137-144) was found in relapsed/refractory MCL patients

Response: We checked the publications from which this information is obtained. The SMARCA4 mutation is found in both untreated and relapsed/refractory MCL patients. (PMID: 24682267, PMID: 30455436). The text has been modified accordingly (165-166).

It is unclear which cancer is discussed in sentences 215-218.

Response: The study was performed using both solid tumors and hematological cancer. The text has been modified to clarify this (250-251).

What is the role of MLL2 mutation in MCL? (222-226)

Response: The role of MLL2 mutation in MCL development and progression is not well understood. However, MLL2 is one of the frequently mutated genes in several B-cell malignancies, including MCL. Interruption of MLL2 function enhances B-cell proliferation and promotes lymphomagenesis. We have modified the text to make this point clear and we have also added a new reference (lines 256-258).

It is unclear which cancer is discussed in sentences 282-285.

Response: The type of cancer has now been mentioned in the text (319).

Round 2

Reviewer 2 Report

The authors have done a nice job of responding to reviewer comments.